# Simvastatin Induces In Vitro Mineralization Effects of Primary Human Odontoblast-Like Cells

**DOI:** 10.3390/ma13204679

**Published:** 2020-10-20

**Authors:** Martin Mariano Isabelo Sabandal, Edgar Schäfer, Jessica Imper, Susanne Jung, Johannes Kleinheinz, Sonja Sielker

**Affiliations:** 1Central Interdisciplinary Ambulance in the School of Dentistry, University of Münster, 48149 Münster, Germany; eschaef@uni-muenster.de (E.S.); jessica-riemp@gmx.de (J.I.); 2Department of Cranio-Maxillofacial Surgery, University Hospital Münster, 48149 Münster, Germany; susanne.jung@ukmuenster.de (S.J.); johannes.kleinheinz@ukmuenster.de (J.K.); sonja.sielker@ukmuenster.de (S.S.)

**Keywords:** mineralization, odontoblast-like cells, pleiotropic effects, simvastatin

## Abstract

Simvastatin (SV) is an often prescribed statin reducing the LDL-concentration in circulating blood. The aim of this study was to evaluate the pleiotropic effects of SV to primary human odontoblast-like cells. Twenty four wisdom teeth of different subjects were extracted and the pulp tissue was removed and minced under sterile conditions. After mincing, the requested cells were passaged according to established protocols. Osteoblastic marker (ALP conversion), viability and mineralization were determined at days 14, 17 and 21 after simvastatin exposition (0.01 µM, 0.1 µM, 1.0 µM, 2.0 µM). The sample size per group was 24 cultures with three replicates per culture for ALP-conversion and mineralization and 6 replicates for viability. A Kruskal–Wallis test was used for statistical analysis. After adding SV, viability was significantly (*p* < 0.01) decreased in a time- and dose-dependent manner, whereas after 21 days, mineralization was significant (*p* < 0.01). ALP-conversion in groups with SV concentrations of 1 and 2 µM SV was significantly (*p* < 0.01) increased. Pleiotropic effects regarding mineralization in higher SV concentrations were possibly induced via alternative mineralization pathways as almost equal elevations of ALP conversion were not evident in the control and experimental groups.

## 1. Introduction

### 1.1. General Function of Statins

Beside others, increased lipid levels in blood could be responsible for first or recurrent coronary events [1]. Since the late 1980s, statins have been widely used as the standard treatment for hypercholesterolemia to reduce low density lipoproteins (LDL) in the blood [2,3]. The formerly used drugs to lower the blood concentration of triglycerides were the so-called fibric acid derivatives (fibrates) but the fibrates lowered the blood concentration of LDL only to a small extent [4].

Simvastatin (SV) is a member of the group of statins, the whole group of the statins is characterized to act as a 3-hydroxy-3-methylglutaryl coenzyme A (HMG-CoA) reductase inhibitor [3], which lowers the circulating LDL concentration in the blood [3]. Out of the group of statins, Simvastatin (SV) is one of the first HMG-CoA reductase inhibitors [3]. Over the years, different types of statins have been used according to the diagnosis. Regarding the often-reported rhabdomyolysis, a statin-related dose-dependent increased risk was supposed so the recommended maximum dose of 80 mg per day was reduced to 40 mg per day by the American Heart Association (ACC/AHA) in 2013 [5].

Within the mevalonate pathway, as a part of the cholesterol biosynthesis, SV inhibits the HMG-CoA in a reversible manner. As a result of the inhibition, the intracellular mevalonate concentration, which is a regulator for HMG-CoA reductase, is reduced. Furthermore, the LDL receptor genes are upregulated, which leads to an increased intracellular uptake of LDL from the bloodstream, resulting in a decrease of LDL concentrations in circulating blood [6].

### 1.2. Formation of Dental Hard Tissues

The formation of dental hard tissues like enamel and dentin with their highly organized and mineralized content of hydroxyapatite crystals is a rigorously controlled process [7,8,9]. The mineralization process of enamel is an epithelial-derived process entirely guided under genetic instruction [10]. On the contrary, the dentin-forming cells are principally derived from connective tissues with an increased mineralized portion [9]. Different hereditary and non-hereditary alterations of dental hard tissues regarding the formation of enamel and dentin have been formerly described [11,12,13].

Due to pathologic alterations during mineralization in the human body, similar mineralization errors within mineralization sequences can lead to similar structural defects in other hard tissues of the body, i.e., subjects with osteogenic defects can also show defects within dentin and vice versa [14]. Compared to dentin, the bone has a continuous, variable equilibrium of resorption and apposition [14]. Once the mineralization of dentin is initiated in utero it does not show resorptive remodeling processes similar to bone and in consequence a continuous apposition process of dentin is evident [14,15]. The apposition of dentin occurs in an incremental manner [16]. Upon dissolution of the basal lamina between ameloblasts and odontoblasts, the mineralization of dentin is initiated, collagen fibrils originated from the odontoblasts connect to the apical surface of the ameloblasts [10]. After initialization of dentin formation, collagenous fibers extend from the centripetal process of the odontoblasts to the dentino-enamel junction [16]. Prior to mineralization, predentin is secreted. After apposition of the minerals, the processes of the odontoblasts retreat and the procedure is repeated by further secretion of predentin and mineralization to follow [10]. In general, the mineralization process is initiated after a wide zone of predentin has been elaborated, the predentinal matrix mainly consists of type I collagen fibrils [15]. Upon the mineralization process, mineralization cores emerge within and in-between the collagenous fibres [16]. Aging and continuous apposition of dentin by the odontoblasts results an increased sclerosis of the dentin and obliteration of the dentinal tubules is possible [17]. However, the formation and composition of enamel differs from that of bone and dentin. The mineral portion of enamel consists mostly of hydroxyapatite crystals with no portion of collagen fibres or other proteins. Amelogenesis as well as dentinogenesis are a genetically controlled process [10].

### 1.3. Pleotropic Effect of Statins

So called pleiotropic effects of statins have been recognized over the years of administration. Due to reported adverse effects, some studies investigated such effects in the cells of other tissues than those originally targeted. Different human cell types had been investigated, such as adipose tissue cells [18,19], osteoblasts [20,21,22,23] and bone marrow cells [24].

Especially the effect of simvastatin upon mineralization was recognized in observational studies, which showed the beneficial effect of statins on increased bone mineral density (BMD) in subjects affected by osteoporosis [25]. Compared to bisphosphonates, statins also influence the mevalonate pathway by inhibiting the HMG-CoA reductase [25]. Beside these observational studies, both in vitro [23,24,26,27] and in vivo studies [28] proved mineralization effects of SV to be bone-related cells like osteoblasts. Besides the typical bone-related cells, other mineralizing cells like human odontoblast-like cells or dental pulp cells were also investigated [29,30,31,32,33,34,35,36,37]. Furthermore, in vivo studies with primary molars [38] have been conducted to investigate possible unknown side effects of SV. SV shows pleiotropic effects like increased odontoblastic differentiation [29,30,31,32,34], increased mineralization [30,32,34,35,36,37], proliferation of odontoblasts [30,32,37] and induction of angiogenesis [31].

The aim of the study was to investigate the influence of SV on mineralization capability and its further influence on the viability of primary human odontoblast-like cells.

## 2. Results

### 2.1. Investigation of the Cytotoxic Concentrations of SV

Prior to the live/dead staining, defined SV concentrations (0.01 µM, 0.1 µM, 1 µM, 10 µM and 20 µM) were added to evaluate toxicity limits of the primary human odontoblast-like cells upon SV exposition. For evaluation of the toxicity limits, two cultures with three replicates were used. The staining was done 24, 48 and 72 h after the beginning of the cell culture. SV concentrations of 10 µM and 20 µM showed a strongly decreased number of vital cells after 48 h. SV concentrations of 0.1 µM and 1 µM showed cytotoxic effects, but the treated cells survived with a reduced viability level (Figure 1). According to the findings of the live/dead staining, the highest used SV concentration was set to 2 µM due to the survival of the cells on a reduced level of viability. The concentration was slightly above the concentration of 1 µM SV, which was evaluated within the cytotoxicity test and markedly lower than the next higher concentration of 10 µM SV.

### 2.2. Cell Viability (MTT-Assay)

Viability of the odontoblast-like cells was significantly decreased (*p* < 0.01) during every point of investigation (day 14, 17 and 21) comparing the group with concentrations of 2 µM and 1 µM SV to the groups with 0.01, 0.1, 1 µM SV and the control group (Figure 2). A comparison between the groups with 0.01 and 0.1 µM SV and the control group over the investigation period revealed a significant (*p* < 0.05) increase of viability from day 14 to day 17 and from day 14 to day 21, whereas the group with 1 µM SV only showed a significant (*p* < 0.05) increase of viability from day 14 to day 21. The results of the 2 µM SV groups showed a significant decrease (*p* < 0.05) from day 14 to day 21.

### 2.3. Osteogenic Marker (ALP-Assay)

ALP-conversion was significantly increased (*p* < 0.01) at every point of investigation (day 14, 17 and 21) comparing the odontoblast-like cells of the groups with 1 µM and 2 µM SV to the groups with 0.01, 0.1, 1 µM SV and the control group (Figure 3). The differences between the groups with 1 µM and those with 2 µM SV were also significant (*p* < 0.01). A comparison of the intragroup values over the investigation period revealed no significant differences (*p* > 0.01).

### 2.4. Mineralization (Alizarin Red S Staining)

The 50% quartile of the groups with 1 and 2 µM SV was increased compared to the groups with 0.01, 0.1 µM SV and control group on day 14. The median value of the 2 µM SV group was significantly (*p* < 0.01) increased compared to the group with 1 µM SV, but not significantly increased compared to the 0.01, 0.1 µM SV groups and the control group. Compared to the median of the control group and the groups with 0.1 and 2 µM SV, the median of the 1 µM SV group was significantly (*p* < 0.01) decreased (Figure 4). Discordant values of an equal range were found in all groups.

The group with 2 µM SV shows on day 17 a significantly increase (*p* < 0.01) in mineralization compared to the groups with 0.01 and 0.1 µM SV and the control group. The range of the 50% quartile of the groups with 1 and 2 µM SV was clearly increased compared to all other groups at day 17. The median value of the 2 µM SV group (0.366) was 6.7 times elevated, whereas the group with 1 µM SV (0.273) showed a 4.96 times increase compared to the control group (0.055). A dose-dependent increase of the median value was recognizable. Discordant values of equal ranges were found in all groups (Figure 4).

The results of day 21 were comparable to those of day 17. Compared to the control group and the groups with 0.01 and 0.1 µM SV, the range of the 50% quartile of the groups with 1 and 2 µM SV was only slightly wider. The median of the group with 2 µM SV was significantly (*p* < 0.01) increased compared to the other groups. The median of the group with 2 µM SV was 1.2 times (1 µM SV) to 2.9 (control) times increased compared to the other groups. The median value of the group with 1 µM SV was also significantly (*p* < 0.01) increased compared to the groups with 0.01 and 0.1 µM SV and the control group, whereby the increase ranged from 1.85 times (0.01 µM SV) up to 2.45 times (control). During the investigation period, a significant (*p* < 0.01) increase within every group over time was recognizable (Figure 4 and Figure 5).

## 3. Discussion

Since 2013, the recommended daily dose of SV ranges from 5 to 40 mg per day [5]. The biological availability is 5% and this corresponds to a systemic available concentration of SV ranging from 0.05 µM to 5 µM. Based on the performed cytotoxicity test in the present study, the chosen SV concentrations were 0.01, 0.1, 1 and 2 µM, respectively. This corresponds well with the SV concentrations used in other studies (0.01, 0.1, 1 and 10 µM) investigating human odontoblast-like and dental pulp stem cells [29,30,31,32,33,34,35,36,37,38,39]. The mentioned studies examined different parameters of human cells, for instance expression levels of ALP-conversion rates and dentin matrix acidic phosphoprotein 1 (DMP-1) [29,30,31,35,36], dentin sialophosphoprotein (DSPP) [29,32,34,35,36,37,39], bone morphogenic protein 2 (BMP-2) [34,36,37,39], osteocalcin (OCN) [30,31,32,34,36,37], osteopontin (OPN) [30,31], osteonectin (ON) [29,31], Runx-2 [30,34,37], platelet endothelial cell adhesion molecule (PECAM-1) [31], mineralization [30,31,34,35,36,37,39] and viability [29,33,34,35,36,37]. For these investigations, especially dental pulp cells [29,31,33,35,36,37,39], dental pulp stem cells [32,34], immortalized dental pulp cells [30] were used. Moreover, one in vivo study of carious primary molars [38] is available.

Several studies investigated the effects of simvastatin on human osteoblasts [20,23,24,40]. Influence of SV on mineralization, alteration of the expression of osteogenic markers and viability were shown but the toxicity levels of osteoblasts were in general lower compared to those of odontoblast-like cells [20,24,39].

In the present study, the influence of higher concentrations of SV on primary human odontoblast-like cells regarding viability, conversion of ALP and mineralization was investigated. Therefore, 24 cultures per group with 3 replicates and 6 replicates per culture were established. The number of investigated primary cultures of dental cells in previous studies ranges from one culture [35], 4 cultures [33,39], 5 cultures [32] to about 90 individual carious primary teeth as part of an in vivo study [38]. In about 50% of the studies the quantity of cultures was not mentioned [29,31,34,36,37]. Thus, in the present study, both the number of cultures (n = 24) and the SV concentration (2 µM) were higher compared to most previously published studies. The only study investigating a larger pool (n = 90 separated in 3 groups) [38] was an in vivo study in which histologic examinations of the pulp tissues after exposition to SV and extraction of the teeth due to orthodontic reasons were performed. However, it should be taken into consideration that the study of Asl Aminabadi et al. [38] examined the effect of SV on an assumed inflammatory stage of deciduous dental pulps.

The overall investigation period in comparable previous studies ranged from 72 h [33], 5 days [32], 14 days [29,31,34], 21 days [35,36,37], 28 days [39] up to 7.41 months in the mentioned in vivo study [38]. The study of Soares et al. [39], with an investigation period of 28 days investigated the long-term effect of SV impregnated scaffolds on dental pulp cells in an artificial inflammatory environment. Hence, the observation period of the present study (21 days) represents the upper range of previously used observation periods.

According to the methodology of other studies [29,33,34,35,36], the viability of the primary odontoblast-like cells after exposition to different SV concentrations was determined in the present investigation. The observation period of other studies regarding the viability of dental cells last up to 14 days [35], while the current study investigated the viability of the cells up to day 21, so the comparability of the results is limited. The results of the current study show an inhibitory effect of SV on the viability of odontoblast-like cells, which was already assumed by Saewong et al. for SV concentrations of 1 µM SV [33]. However, the limiting concentration was found to be 2 µM SV (Figure 2). In agreement with other studies [29,31,33,35], the present results revealed a significant (*p* < 0.01) decrease of cell viability in the groups of 1 and 2 µM SV compared to the control and the 0.01 and 0.1 µM groups. Especially the group with 2 µM SV showed a significant (*p* < 0.01) decrease of viability at each examination point. The groups with 1 and 2 µM SV showed a marked reduction of the 50% quartiles compared to the other groups (Figure 2). This can be explained by the cytotoxic effects of SV, which can contribute to a significant limitation of the metabolism of the cells, in a consequence the viability is correspondingly limited. The median values and the 50% quartiles of all groups displayed a slight elevation at day 21 compared to day 14 (Figure 2). In summary, with regard to the viability of odontoblast-like cells it can be concluded that more pronounced cytotoxic effects of SV can be expected upon concentrations higher than 1 µM SV. An effect on the viability of odontoblast-like cells due to the use of ethanol (abs) as a solvent cannot be completely excluded. However, the maximum dilution used at the SV concentration of 2 µM corresponds to a dilution ratio of 1:500 (0.5%). Furthermore, ethanol is a physiological metabolite in the cell metabolism of odontoblast-like cells and is therefore found in the cytosol of the examined cells.

The determination of ALP conversion is often used as a typical odontogenic [29,30,31,35,36,39] and osteogenic marker [41,42] during odontogenic [29,31,34,35,36,37,39] and osteogenic differentiation [41]. The conversion of the ALP is usually elevated during biomineralization processes [43]. Additionally, a slight ALP activity was found within inflammatory free healthy dental pulp tissue but the ALP level increase when the pulp status switched to an inflammatory stage, in as far as an increased mineralization with an increased ALP activity was found [44]. Different types of ALP are known in human beings but the tissue-nonspecific alkaline phosphatase (TNAP) is essential for biomineralization in bone and other mineralized tissues [43]. Under the influence of SV, an increased ALP conversion is a possible sign of enhanced biomineralization by the odontoblast-like cells. This effect was already reported [29,30,35,36,39] for SV concentrations up to 1 µM SV, which is lower than the highest SV concentration used in the present study. Only one study investigated the influence of 10 µM SV [31]. So far, the investigated concentrations of other studies ranged from 0.01 µM SV [29,31,35], 0.1 µM SV [31,35,39], 1 µM SV [30,31] up to 10 µM SV [31]. The results of these studies showed increased ALP-conversion values at different investigation points. Only 2 studies [35,39] extended the observation period up to 21 days, but the used SV concentration was limited to a maximum concentration of 0.1 µM SV.

The conversion of ALP relative to the total expressed protein content was determined in the current study. A time- and dose-dependent increase of the ALP-conversion is shown in the current study upon the influence of SV. During every investigation point, the groups with 1 µM and 2 µM SV showed significantly (*p* < 0.01) elevated levels of ALP-conversion compared the control group and the groups with 0.01 and 0.1 µM SV with significant differences between 1 µM and 2 µM SV. Compared to other studies, the present results confirm the increase of ALP-conversion especially for higher concentrations (Figure 2). Although viability dropped significantly (*p* < 0.05) during the investigation period in the 2 µM SV group, there was almost no change in ALP-conversion over time.

The significant increase of the ALP-conversion of the groups of 1 and 2 µM SV combined with the findings of a significant dose-dependent decrease of the viability of the odontoblast-like cells implies a marked effect of SV on the mechanism of biomineralization of primary odontoblast-like cells. In summary, the present results corroborate those of Min et al. [31], who suggested a time and a dose-dependent increase of the ALP-conversion under the influence of SV.

Based on a previously published protocol, the influence of SV on the mineralization of odontoblast-like cells was investigated in this study [45]. Frequently, the influence of SV on biomineralization was investigated by alizarin red S staining and subsequent redissolution with determination of photometric absorbance [29,31,34,35,36,37,39]. According to the results of the present study, the 50% quartiles of the groups of 1 and 2 µM SV were markedly increased compared to the control, the 0.01 and 0.1 µM SV groups. Additionally, the median value of the 2 µM SV group was clearly elevated compared to the control, the 0.01, 0.1 and 1 µM SV groups (Figure 4). Thus, at day 17 and 21, the groups with higher SV concentrations of 1 and 2 µM SV displayed an increased mineralization compared to all other groups. These findings agree with those of other studies in as far as higher concentrations of SV [31,34,35,37] resulted in more pronounced mineralization capabilities.

In summary, the findings of the present study showed a dose-dependent decrease of the cell viability under the influence of SV. The results are consistent with the findings of previously published studies [29,33,35,36]. In contrast to the often shown decrease of cell viability, Zijah et al. and Samiei et al. [34,37] reported an increase of the viability of odontoblast-like cells under the influence of scaffolds impregnated with SV. However, the use of scaffolds can alter the influence of SV to odontoblast-like cells. Although the viability of the odontoblast-like cells was significantly reduced especially with higher concentrations of 1 and 2 µM SV, the ALP-conversion in these groups was significantly increased (*p* < 0.05) over time. Significantly decreased cell viability, increased phosphate turnover and significant (*p* < 0.01) increase of the mineralization capability of the odontoblast-like cells point to a linkage between SV and the ability of biomineralization of odontoblast-like cells especially at a later stage of culture in the current study (day 21) (Figure 4) [30,34,35,37]. Although cytotoxic effects of SV are recognizable in a time and dose-dependent manner upon human osteoblasts, odontoblast-like cells and likewise other cell types [23,24,46,47] the application of SV induced a 2 times elevated median value of the ALP conversion in the group with 2 µM SV compared to the control group (Figure 3). Additionally, the median value of the mineralization capability of the group with the same concentration was approximately 3 times higher compared with the control group (Figure 4). The findings regarding ALP-conversion, mineralization capability and extremely reduced viability (control group approximate 10 times higher) in the group with 2 µM SV (Figure 2) implies a strong influence of SV on mineralization of mineralizing cells like the odontoblast-like cells and a possible alternative biomineralization pathway induced by SV.

## 4. Materials and Methods

### 4.1. Ethics Approval and Study Design

The present study investigated the effects of SV on 24 primary human odontoblast-like cell cultures of different human donors ranging in age from 18 to 35 years. Extraction of the impacted and displaced third molars of all subjects was performed due to a lack of space within the dental arch. Analysis of mineralization, cell viability and effects on osteogenic marker (ALP) was performed. According to the “Declaration of Helsinki” the study was designed and approved by the Ethics Committee of the Faculty of Medicine, University of Muenster (#2016-443-f-S). Written informed consent was obtained from all donors prior to the extraction and cell isolation.

Exclusion criteria of the study:tumors in the head and neck area;previous systemic administration of statins;subjects younger than 18 years;possible pregnancy of the subjects.

### 4.2. Obtaining the Dental Pulp Samples

The dental pulp was removed anonymously and under sterile conditions after extraction. Human caries free third molars were collected after extraction in the Department of Cranio-Maxillofacial Surgery, University Hospital Münster (Münster, Germany). After disinfection of the root surfaces, the periodontal tissues were removed by curettage and the tooth surface was rinsed with alcohol (70% 2-Propanol, Carl Roth, Karlsruhe, Germany). To gain access to the pulp cavity, the teeth were opened by sterile cylindrical diamond burs at the enamel-cement boundary. In the following the crown was fractured by a widening of the resulting slot with crown spreading forceps (Stoma Dentalsysteme, Emmingen-Liptingen, Germany). After opening of the pulp cavity, the containing tissue was removed with sterile Hedstroem files ISO-sizes 10 and 15 (VDW, Munich, Germany).

### 4.3. Cultivation and Isolation of the Primary Human Odontoblast-Like Cells

As described previously, the isolation and culture techniques of primary human cells were performed [45,48]. The pulp tissue was minced under sterile conditions and the tissue fragments were fixed to the dry surface of cell culture dishes by adhesion for a short time under a sterile bench. To avoid mechanical disturbances, culturing medium was added carefully to fixed pulp particles in the cell culture dishes. After the cells grew out of particles, these particles were removed after 14 days. Cultivation of the isolated cells were done in Dulbecco’s Modified Eagle Medium (DMEM) low glucose (Gibco, Dreieich, Germany) with supplement of 10% bovine calf serum, 1% Amphotericin B (250 mg/mL) and 1% Penicillin (10.000 U/mL)/Streptomycin (10.000 g/mL) (all Biochrom, Berlin, Germany). Cultivation of the cells was done at 37 °C in a humidified atmosphere with 5% CO_2_. Replacement of the culturing medium was done every two to three days and when reaching 90% of confluence, the cells were passaged. Mineralization was induced by adding 16 ng/mL dexamethasone (Fortecortin, Merck Pharma, Darmstadt, Germany), ascorbic acid (1.4 mM) and ß-glycerophosphate (10 mM) (all Sigma-Aldrich, Hamburg, Germany). SV was dissolved in ethanol (abs) to a final stock solution of 6 mM and 1 mM stored at 4 °C (Sigma-Aldrich, Hamburg, Germany). Various SV concentrations (0.01 µM, 0.1 µM, 1.0 µM and 2.0 µM) were prepared by diluting a stock solution of SV with the culturing medium. Final concentration of ethanol was according to the dilution 0.2% in the culturing medium of 1 µM SV.

Density of the cells was 5000 cells/cm^2^ in 48-well plates after seeding (Greiner Bio-One, Frickenhausen, Germany). The cells were allowed to adhere for 24 h prior to the adding of SV of defined concentrations to the culturing medium (0.01 µM, 0.1 µM, 1.0 µM, 2.0 µM) after inducing the mineralization. Culturing medium with SV was replaced every two to three days throughout the cell culture study.

Cells with osteogenic induction and without SV served as a control group. Cell viability and osteogenic activity and mineralization capability were analyzed at days 14, 17 and 21. The sample size for each group including the control group for mineralization capability and ALP-conversion was 24 cultures with three replicates whereas the sample size for viability groups including the control group was 24 cultures with 6 replicates each. Control groups were cells in culture without SV.

### 4.4. Characterization of the Primary Odontoblast-Like Cells

Primary human odontoblast-like cell cultures were characterized by identification of dentine sialophosphoprotein (DSPP) gene product by PCR following agarose gel electrophoresis. Primers are listed in Table 1. For RNA isolation and purification, a RNeasy Kit (Qiagen, Hilden, Germany) was used and performed according to the manufacturer’s protocol. Purity and concentration of isolated RNA was determined by a spectrophotometric reading (NanoDrop™ 2000, ThermoFisher Scientific, Wesel, Germany). RNA was transcribed in complementary DNA with the MMLV Reverse Transcriptase 1st-Strand cDNA synthesis Kit (Epicentre, Madison, WI, USA) according to manufacturer’s protocol. Upstream, a DNase treatment to eliminate existent genomic DNA was performed (Baseline-Zero DNase, Epicentre, Madison, WI, USA). cDNA was amplified with Eppendorf Mastercycler (Eppendorf, Hamburg, Germany) using PCR Master mix (2x) (ThermoFisher Scientific, Wesel, Germany). PCR protocol according to manufacturer’s protocol.

### 4.5. Determination of the Nontoxic Simvastatin Concentration

Odontoblast-like cell were seeded with a density of 10,000 cells/cm^2^ in 48-well plates (Greiner Bio-One International GmbH, Kremsmünster, Austria). Before SV freshly diluted in culturing medium (0.01 µM up to 20 µM) was added the cells were allowed to adhere for 24 h. Cells without added SV in culturing medium served as a control group. Cell viability was analyzed 24 h, 48 h and 72 h after adding SV.

### 4.6. Cell Viability (MTT-Assay)

An in-house MTT assay was used to estimate cell viability. The cellular NAD(P) reflux converts the yellow thiazolyl blue tetrazolium bromide (0.5 mg/mL) (Sigma-Aldrich, Hamburg, Germany) into the violet formazan dye, which is measured photometrically at 570 nm wavelength. Pierce™ LDH Cytotoxicity Assay (ThermoFisher Scientific, Wesel, Germany) was used to determine cytotoxic effects (data not shown). All assays were performed according to the manufacturers’ protocols. Colorimetric determination was done with the µQuant™ reader (BioTek, Winooski, VT, USA).

The qualitative analysis of cell viability was performed using fluorescein diacetate (FDA)/propidium iodide (PI) staining. Viable cells were stained green by FDA (Sigma-Aldrich, St. Louis, MO, USA) and nuclei of necrotic and apoptotic cells were stained red by PI (Fluka, Sigma-Aldrich, St. Louis, MO, USA).

### 4.7. Activity of Alkaline Phosphatase

As an activity of alkaline phosphatase, the ALP-conversion relative to the total protein was determined. For both protein expression analysis and ALP-conversion, cells were lysed with the Pierce™ IP Lysis Buffer (ThermoFisher Scientific, Wesel, Germany). The supernatant was frozen at −80°C for subsequent assays. Pierce™ BCA Protein Assay (ThermoFisher Scientific, Wesel, Germany) was used for protein quantification. The Alkaline Phosphatase Assay Kit (abcam, Cambridge, UK) was used for detection of alkaline phosphatase activity. The necessary total protein expression was determined according to the methods described in the studies of Liu et al. and Zhao et al. [22,46] All assays were performed according to the manufacturers’ protocols. Colorimetric determination was done with the µQuant™ reader (BioTek, Winooski, VT, USA).

### 4.8. Mineralization Capability

A modified alizarin red S staining (Alizarin Red S Staining Quantification Assay, ScienCell, Carlsbad, CA, USA) was used for determination of the mineralization in cell culture [49]. After fixation with formaldehyde (4% in phosphate buffered saline) the culture was stained with an alizarin red S solution (40 nM, pH 4.1). Photographs were made for documentation (Figure 5). The stained cells were lysed in 10% acetic acid, heated to 85 °C for 10 min and centrifuged for 10 min at 20,000 × g prior to quantification. A 10% ammonia solution was used to neutralize the supernatant. The measurement of the dissolved alizarin red S was done at wavelengths of 405 nm. Colorimetric determination was done with the µQuant™ reader (BioTek, Winooski, VT, USA).

### 4.9. Statistical Analysis

The statistical software SPSS version 26 (IBM, Ehningen, Germany) was used for statistical analysis. For each observation day, the results of the corresponding samples were assigned to the specified groups. Due to the lack of normal distribution and small sample sizes, statistical analysis was done by using the Kruskal–Wallis test. Significance levels were set at *p* < 0.05.

## 5. Conclusions

Upon the influence of SV, a time- and dose-dependent decrease of viability and a significant increase of both ALP turnover and mineralization was recognizable. The present study confirms a higher tolerance of odontoblast-like cells to higher simvastatin concentrations as are hitherto known.

Pleiotropic effects for SV could be detected earlier, but a deeper insight into the biomineralization processes of human cells should be gained in this respect. Besides the direct influence of SV to cell cultures is also the influence of scaffolds [34,37], which in combination with SV seems to be an interesting field for further investigation. Studies with or without scaffolds, regarding other treatment options like pulp capping agents, regenerative endodontic treatments and the influence of systemically administered SV and/or other statins may be topics to further elucidate the pleiotropic effects of SV with regard to possible clinical applications.

## Figures and Tables

**Figure 1 materials-13-04679-f001:**
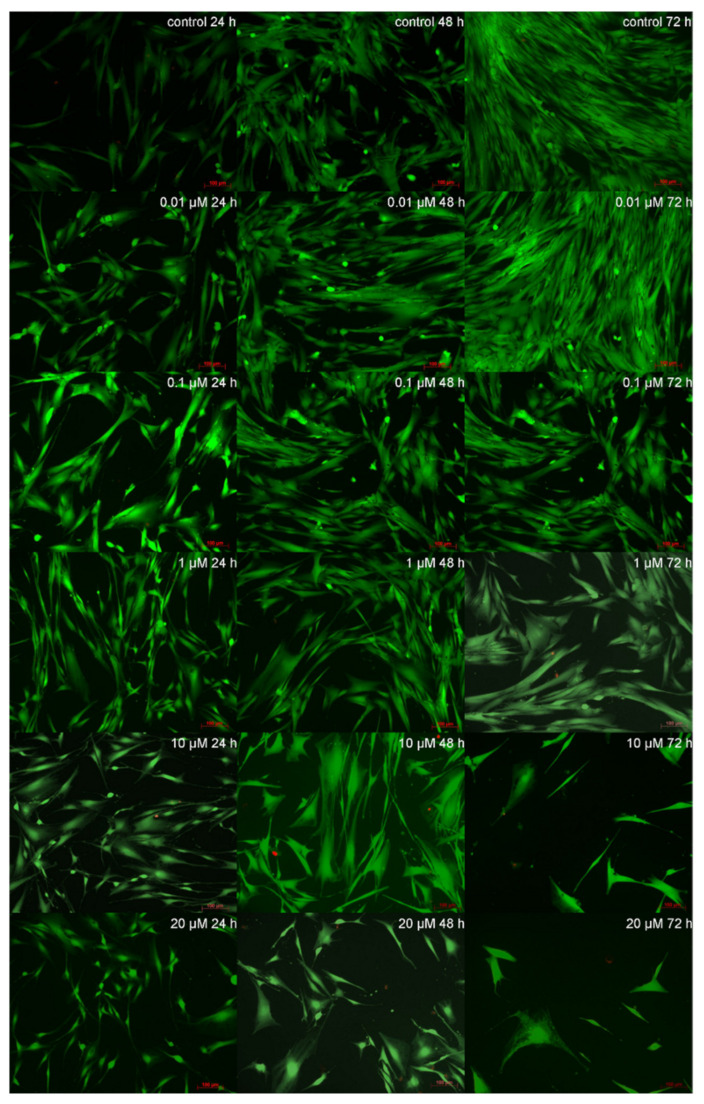
Live/dead staining: *x*-axis time, *y*-axis downwards ascending concentrations of added simvastatin (SV); viable cells green, necrotic and apoptotic cell nuclei red. (magnification ×100; scale bar 100 µm (in red at the lower right corner of each picture)).

**Figure 2 materials-13-04679-f002:**
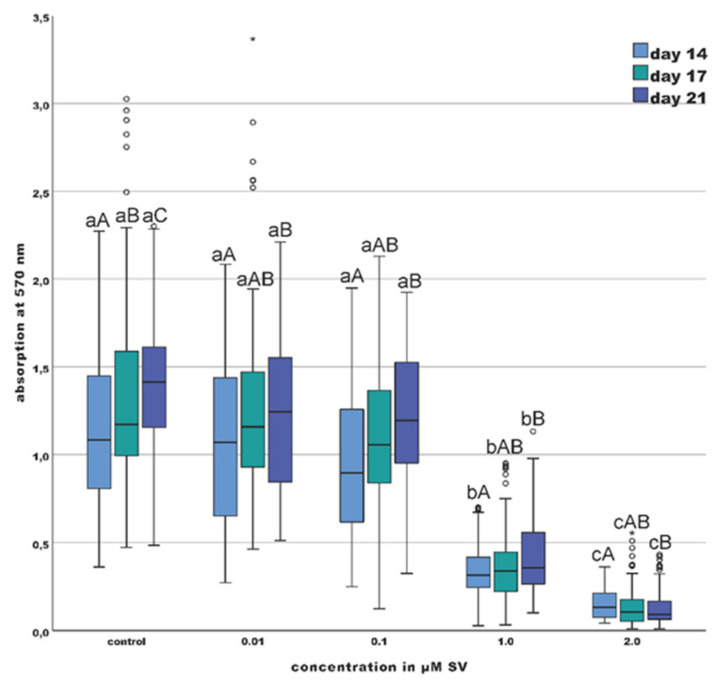
Viability given as photometric absorption at λ 570 nm of the MTT-assay; (different letters indicate statistically significant differences at *p* < 0.05 within the day (capitalized) and concentration (uncapitalized)); (the upper and lower whiskers show the maximum and minimum values, the box represents 50% of the values and the horizontal line within the box displays the median value).

**Figure 3 materials-13-04679-f003:**
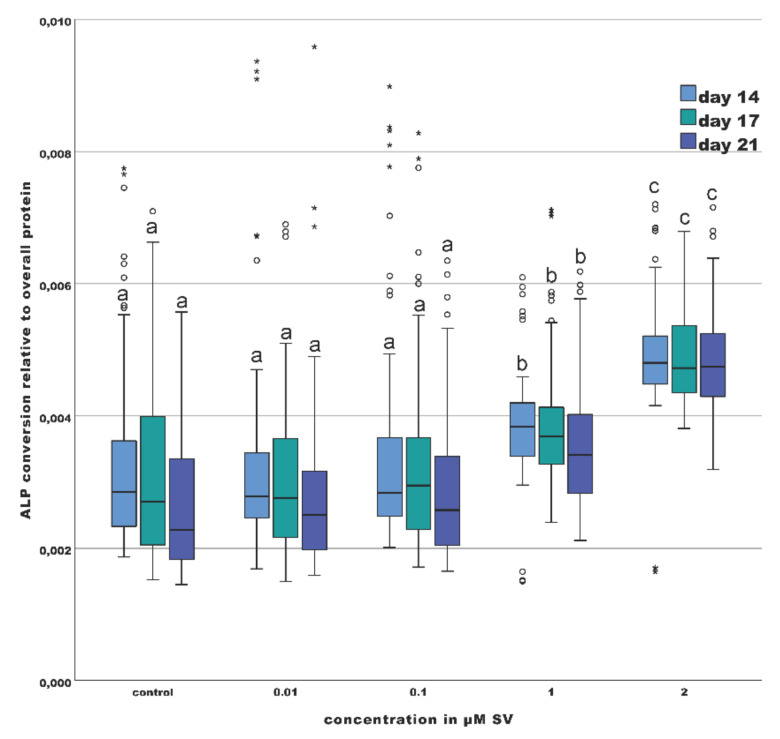
Conversion of alkaline phosphatase (ALP) normalized to the overall protein; (different letters indicate statistically significant differences at *p* < 0.05); (the upper and lower whiskers shows the maximum and minimum values, the box represents 50% of the values and the horizontal line within the box displays the median value).

**Figure 4 materials-13-04679-f004:**
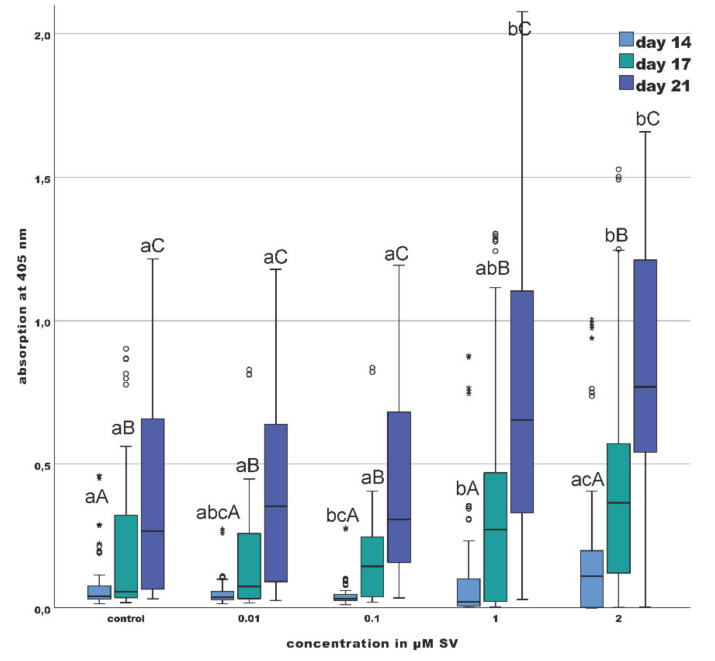
Mineralization given as photometric absorption at λ 405 nm of the alizarin red S staining; (different letters indicate statistically significant differences at *p* < 0.05 within the day (capitalized) and concentration (uncapitalized)); (the upper and lower whiskers shows the maximum and minimum values, the box represents 50% of the values and the horizontal line within the box displays the median value).

**Figure 5 materials-13-04679-f005:**
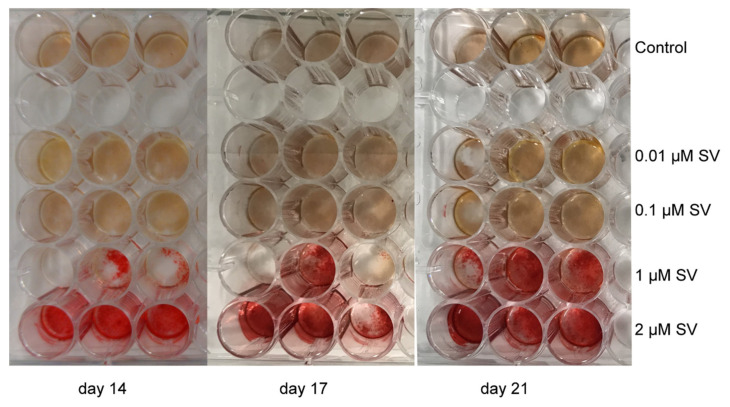
Alizarin S red staining prior to the dilution. From left to right ascending time in days; top row control and below the ascending concentrations as marked on the right side.

**Table 1 materials-13-04679-t001:** Primers used in PCR.

Gene	Primer Forward	Primer Reverse	Product Length	Annealing Temperature
DSPP	GTCGCTGTTGTCCAAGAAGA	ATCCTCATCTGCTCCATTCC	239 bp	63 °C
GAPDH	CTCAGACACCATGGGGAAGG	TCGCTCCTGGAAGATGGTGA	249 bp	54 °C

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
