# Peer review of "Simvastatin Induces In Vitro Mineralization Effects of Primary Human Odontoblast-Like Cells"

_materials, 2020, doi:10.3390/ma13204679_

Round 1

Reviewer 1 Report

In line 81, the authors compare the results of 2 μM and 1 μM with 0.01 μM and 0.1. μM. The concentration of 1 μM is repeated incorrectly.

In line 252 the topic should be Material and Methods and not Results

The authors assess the influence of SV on mineralization capability and viability upon primary human odontoblast-like cells. The plan and methods of the project are adequate, the results are adequately described and presented.

The discussion focus very well the topic and make a complete comparison with the state of the art.

The conclusions are consistent with the obtained results and pave the way to a better comprehension of other effects of simvastatin with potential relevant clinical impact. I clearly recommend the publication of the paper.

Author Response

Answers to Reviewer 1:

In line 81, the authors compare the results of 2 μM and 1 μM with 0.01 μM and 0.1. μM. The concentration of 1 μM is repeated incorrectly.

Within the investigation of the toxicity limits defined concentrations of SV were used. As the staining shows that 1µM SV causes a reduced viability level of the odontoblast-like cells, a concentration (2µM SV) slightly above 1µM SV was chosen. The paragraph has been altered.

“According to the findings of the live/dead staining the highest used SV concentration was set to 2 µM due to the survival of the cells on a reduced level of viability. The concentration was slightly above the concentration of 1µM SV, which was evaluated within the cytotoxicity test and markedly lower than the next higher concentration of 10µM SV.” (line 112-115)

In line 252 the topic should be Material and Methods and not Results

The headings were corrected and the sub-paragraphs have been altered too. (line 299-301, 314, 324, 352, 365, 370, 380, 390, 399)

Reviewer 2 Report

The manuscript is fairly written. However, the claim is not well supported by the obtained data. A similar report has been published by the authors using human primary osteoblasts.

My major concerns are the following:

  1. In Figure 1 the author showed only cell staining images without any qualitative analysis to support the claim written in line 82 „According to the results the highest used SV concentration was set to 2 μM“. This reviewer could not see any qualitative analysis that came to this conclusion. The author should present the dose-response curve and calculate the EC50 from this data. Besides, the author should state the number of experimental replicates. Please also include the scale bar.
  2. In Figure 2 the author utilized MTT-Assay for cell viability test. However, MTT assay is correlated with the metabolic functions of cells. Do the metabolic functions affect the readout? This reviewer would suggest normalizing the data to the control for each donner. It will make the data easier to interpret. Also, from the current data, this reviewer would assume that simvastatin inhibited the proliferation of cells. The author should be careful with the interpretation of MTT data, since it is linked to the ALP assay and also mineralization.
  3. The author mentioned LDH cytotoxicity assay in the method in line 326. However, the author did not provide any data from this assay. This assay is important to support the MTT test and also to clarify whether cells underwent apoptosis or the proliferation is inhibited.
  4. The ALP-Assay is not sufficient to demonstrate the osteogenic marker. At least the author should perform a qPCR analysis for Coll I, Runx2, and OCN.
  5. In Figure 4, the author should show representative images of Alizarin Red S staining to support the qualitative analysis. Also, this reviewer would suggest normalizing the Alizarin Red S results to the number of cells., since the cell number is not the same.
  6. As odontoblast can secret ECM and in turn alter their mineralization, do the authors observe an alteration in ECM?

Minor:

  1. Please check if „1.1. Formation of dental hard tissues“ in line 25 is needed.
  2. Since the experiment was performed for 21 days. When did the author change the cell culture media? This has to be stated in the materials and methods.

Author Response

Answers to reviewer 2:

  1. In Figure 1 the author showed only cell staining images without any qualitative analysis to support the claim written in line 82 „According to the results the highest used SV concentration was set to 2 μM“. This reviewer could not see any qualitative analysis that came to this conclusion. The author should present the dose-response curve and calculate the EC50 from this data. Besides, the author should state the number of experimental replicates. Please also include the scale bar.

In this primary study, we wanted to determine nontoxic simvastatin concentration prior to the definite cell culture study. Simvastatin concentrations showing clearly toxic effects to cell culture in MTT assay, live/dead staining and LDH assay were excluded. In summary, the chosen biologic active simvastatin concentrations which showed nontoxic or just minor cytotoxic effects during the first days of proliferation the concentrations ranges up to 2 µM simvastatin. The ambiguous wording was adjusted accordingly.

“According to the findings of the live/dead staining the results the highest used SV concentration was set to 2 µM due to the survival of the cells on a reduced level of viability. The concentration was slightly above the concentration of 1µM SV, which was evaluated within the cytotoxicity test and markedly lower than the next higher concentration of 10µM SV.” (line 112-114)

The evaluation of all performed assays shows that the chosen simvastatin concentration had no or just slight effects on cell survival. In the consequence, we decided to present the picture of the live/dead staining because these pictures illustrate the results at it bests. The primary study was done with two cell cultures and in three replicates. The information has added to the manuscript.

“For evaluation of the toxicity limits 2 cultures with three replicates were used.” (line 108)

Pictures of the living / dead staining were recorded with a x100 magnification and the scale bar (red bar at the bottom right in each picture) is 100 µm.

“; (magnification x100; scale bar 100 µm [in red at the lower right corner of each picture]).” (line 118-119)

  1. In Figure 2 the author utilized MTT-Assay for cell viability test. However, MTT assay is correlated with the metabolic functions of cells. Do the metabolic functions affect the readout? This reviewer would suggest normalizing the data to the control of each donner. It will make the data easier to interpret Also, from the current data, this reviewer would assume that simvastatin inhibited the proliferation of cells. The author should be careful with the interpretion of MTT data, since it is linked to the ALP assay and also mineralization.

During the MTT assay the yellow thiazolyl blue tetrazolium bromide is added to the cell culture via the cellular redoxpotential (NAD(P)) the bromide is reduced to the purple formazan dye. When the conversion is done the cells were lysed and the adsorption at 570nm wavelength so there is no metabolic function existent to cause any effect on readout.

To present the data we tried several types of graphs to show the data but comparing to each other the present graphs show the findings at best.

We don’t want to link the different assays to each other. As you state the results must be considered relative to each other.

  1. The author mentioned LDH cytotoxicity assay in the method in line 326. However, the author did not provide any data from this assay. This assay is important to support the MTT test and also to clarify whether cells underwent apoptosis or the proliferation is inhibited.

In the preliminary study we analyzed the release of LDH according to cell culture design, in four cell cultures. In these four cell cultures, no noticeable effects could be observed. The release of LDH to culturing media, was always lower compared to control. The diagram below shows the results of one cell culture as an example. During data evaluation, we decided to exclude the LDH assay from study design because of less expressiveness. To avoid an overburden and to keep more clearness, data were not included in the manuscript.

The text of the manuscript was altered as follows:

“Cytotoxic effects were determined with the Pierce™ LDH Cytotoxicity Assay (ThermoFisher Scientific) (data not shown)” (line 371)

  1. The ALP-Assay is not sufficient to demonstrate the osteogenic marker. At least the author should perform a qPCR analysis for Coll I, Runx2, and OCN.

The study design of the present study did not include gene expression analysis. The primary objective of the study was to evaluate the protein levels and/or their analogues. Upon analysis of the data the missing of the specific gene analysis emerges so these investigations will be included within further studies regarding pleiotropic effects of simvastatin.

  1. As odontoblast can secret ECM and in turn alter their mineralization, do the authors observe an alteration in ECM?

Observations regarding secretion of ECM was no part of the present study. As well as gene expression levels also possible alterations of ECM secretion under influence of SV may be of future interest within further studies regarding pleiotropic effects of SV.

            Please check if “1.1. Formation of dental hard tissues” in the line 25 is needed.

The section introduction has been altered and extended, so some 2 other sub-paragraphs have been added.

“1.1. General function of Statins

Beside others, increased lipid levels in blood could be responsible for first or recurrent coronary events [1]. Since the late 1980s the drugs of the group of statins were administered as a common therapeutic agent to reduce the concentration of low-density-lipoproteins (LDL) in blood [2, 3]. The former used drug to lower the blood concentration of triglycerides was the the so called fibric acid derivatives (fibrates) but the fibrates only show little effect on the circulating LDL blood concentration [4].

The group of statins includes Simvastatin (SV), the whole group of the statins is characterized to act as a 3-hydroxy-3-methylglutaryl coenzyme A (HMG-CoA) reductase inhibitor [3], which lowers the circulating LDL concentration in blood [3]. Out of the group of statins, Simvastatin (SV) is one of the first HMG-CoA reductase inhibitors [3]. During the use over years different types of statins are known and used depending on the diagnosis. Upon often reported rhabdomyolysis a statin related dose-dependent increased risk was supposed. In the following, the formerly recommended daily maximum dose of 80mg was reduced in 2013 by the American Heart Association (ACC/AHA) to 40mg per day [5].

The target location of SV is a reversible inhibition of the HMG-CoA reductase within the cholesterol biosynthesis and mevalonate pathway. The inhibition leads to a reduced intracellular concentration of mevalonate which serves as a regulator of the HMG-CoA reductase. Additionally, the expression of LDL receptors is upregulated [6] Due to the upregulation of the LDL receptor expression the cellular intake of LDL from the circulating blood is increased causing a lowering of the LDL within the circulating blood [6].” (line 25-46)

“Especially the effect of simvastatin upon mineralization was recognized in observational studies which showed beneficial effect of statins on increased bone mineral density (BMD) in subjects affected by osteoporosis [25]. Compared to bisphosphonates statins also influence the mevalonate pathway by inhibiting the HMG-CoA reductase [25]. Beside these observational studies, both in vitro [23,24,26,27,] and in vivo studies [28] proved mineralization effects of SV to bone related cells like osteoblasts.” (line 91-96)

Since the experiment was performed for 21 days. When did the author change the cell culture media? This has to be stated in the materials and methods.

The culturing media was changed every two to three days throughout the cell culture study. We mention this in line 296f of materials and methods section.

“Freshly mixed culturing medium with SV was replaced every two to three days throughout the cell culture study.” (line 344-345)

Reviewer 3 Report

In this study material and methods section are titled as results. And again in the next part of this paper, section is titled as results. It is unclear and it is necessary to prepare paper according to instructions for authors.

Is it possible to correlate obtained results to clinical studies.

In material section - It was not clear why staining was done 24, 48 and 72h after beginning of the cell culture. Explain why did you choose such time interval.

Author Response

Answers to reviewer 3:

In this study material and methods section are titled as results. And again in the next part of this paper, section is titled as results. It is unclear and it is necessary to prepare paper according to instructions for authors.

The headings were corrected and the sub-paragraphs have been altered too. (line 299-301, 314, 324, 352, 365, 370, 380, 390, 399)

Is it possible to correlate obtained results to clinical studies.

Except for one study, there is currently no other clinical trial that deals with the effect of statins on odontoblast-like cells. This study investigates the potential correlation of statins and the obliteration of pulp chambers.

In material section - It was not clear why staining was done 24, 48 and 72h after beginning of the cell culture. Explain why did you choose such time interval.

In this primary study, we want to determine nontoxic simvastatin concentration prior to the definite cell culture study which shows the maximum concentration with which the cells can survive. Simvastatin concentrations which show clearly toxic effects to cell culture in MTT assay, live/dead staining and LDH assay were excluded. In summary, biologic active simvastatin concentrations which show nontoxic or just showed minor cytotoxic effects during the first days of proliferation the concentrations ranges up to 2 µM simvastatin. As the LDH assay shows the survival of the cells is evident with the chosen SV concentrations, so it was not necessary to prolong the staining.

The diagram below shows the results of one cell culture as an example. During data evaluation, we decided to exclude the LDH assay from study design because of less expressiveness. To avoid an overburden and to keep more clearness, data were not included in the manuscript.

Reviewer 4 Report

The authors present a scientifically sound and well written, but poorly organized paper on the effect of SV on human OLC's. The organization of the paper should be edited (e.g. Results are mentioned twice, several subparagraphs should be combined / reorganized). 

In the Discussion, the range of literature is confusing and difficult to follow. Try to keep literature down to 3 per brackets (unless absolutely necessary), otherwise it is confusing to the reader and hard to focus on this information. 

Apart from these minor mistakes, this paper is of high quality and should be commended. Please review minor spelling and grammatical errors. 

Author Response

Answers to reviewer 4:

The authors present a scientifically sound and well written, but poorly organized paper on the effect of SV on human OLC's. The organization of the paper should be edited (e.g. Results are mentioned twice, several subparagraphs should be combined / reorganized). 

The headings were corrected and the sub-paragraphs have been altered too. (line 299-301, 314, 324, 352, 365, 370, 380, 390, 399)

In the Discussion, the range of literature is confusing and difficult to follow. Try to keep literature down to 3 per brackets (unless absolutely necessary), otherwise it is confusing to the reader and hard to focus on this information.

We tried to focus on the absolutely necessary references. However, in the cases of more than three references given to support a statement we believe that these all are of relevance for the reader.

Apart from these minor mistakes, this paper is of high quality and should be commended. Please review minor spelling and grammatical errors.

The spelling and grammatical errors have been reviewed and corrected

Reviewer 5 Report

Simvastatin induces mineralization effects of primary human odontoblast-like cells

Materials (ISSN 1996-1944)

MANUSCRIPT NUMBER:  materials-954063

The aim of the present investigation was to to evaluate pleiotropic effects of SV to primary human odontoblast-like cells.

GENERAL COMMENTS

The study is original, but some points of the manuscript should be improved. The following major comments are suggested:

-Title: it should be clarified that the present investigation is an in vitro study (ex: “Simvastatin induces in vitro mineralization effects of primary human odontoblast-like cells”)

- Introduction:

In the section is present only one sub-paragraph. This choice is incomprehensible and it should be deleted.

The statins should be introduced better and thorough about theirs effect on tissue mineralization in order to clarify the rationale of the study. Different previous studies described those effect on bone tissue.

  • Shah SR, Werlang CA, Kasper FK, Mikos AG. Novel applications of statins for bone regeneration. Natl Sci Rev. 2015;2(1):85-99. doi:10.1093/nsr/nwu028
  • Horiuchi N, Maeda T. Statins and bone metabolism. Oral Dis. 2006 Mar;12(2):85-101. doi: 10.1111/j.1601-0825.2005.01172.x. PMID: 16476028.
  • Garrett IR, Mundy GR. The role of statins as potential targets for bone formation. Arthritis Res. 2002;4(4):237-240. doi:10.1186/ar413

Methods

No details about the age range of the donors has been indicated. Why the teeth were extracted? Were they germs, impacted teeth or erupted ones?

Results:

Figure 2,3 and 4 presented too low small fonts and should be improved in quality. The bars and line legends should be described in the captions (standard deviation? Minimum/maximum? Mean/Median?)

No images of alizarin red colture has been added? Why?

Discussion

Line 99: why the author analyzed osteogenic markers? What is the connection with odontoblasts activity? It should be clarified and discussed.

Discussion:

Line 170: The following sentence is a repetition of introduction section. Conometric retention does not incorporate cement or screw retention. The discussion and conclusion provided a limitate overview due to the lack of studies identified with the electronic databases screening. Probably a deeper manual search could be useful to increase the manuscripts eligible for the review in order to give strength to the discussion.

Conclusion: The conclusions are generic and should be incentrated to the high survival rate and prosthetic retention capability of the conometric implant systems emerged by the review.

Author Response

Answers to reviewer 5:

-Title: it should be clarified that the present investigation is an in vitro study (ex: “Simvastatin induces in vitro mineralization effects of primary human odontoblast-like cells”)

The title was altered according to your recommendation.

“Simvastatin induces in vitro mineralization effects of primary human odontoblast-like cells” (line 2)

- Introduction:

In the section is present only one sub-paragraph. This choice is incomprehensible and it should be deleted.

The introduction section has been altered and extended, so some 2 other sub-paragraphs have been added.

“1.1. General function of Statins

Beside others, increased lipid levels in blood could be responsible for first or recurrent coronary events [1]. Since the late 1980s the drugs of the group of statins were administered as a common therapeutic agent to reduce the concentration of low-density-lipoproteins (LDL) in blood [2, 3]. The former used drug to lower the blood concentration of triglycerides was the the so called fibric acid derivatives (fibrates) but the fibrates only show little effect on the circulating LDL blood concentration [4].

The group of statins includes Simvastatin (SV), the whole group of the statins is characterized to act as a 3-hydroxy-3-methylglutaryl coenzyme A (HMG-CoA) reductase inhibitor [3], which lowers the circulating LDL concentration in blood [3]. Out of the group of statins, Simvastatin (SV) is one of the first HMG-CoA reductase inhibitors [3]. During the use over years different types of statins are known and used depending on the diagnosis. Upon often reported rhabdomyolysis a statin related dose-dependent increased risk was supposed. In the following, the formerly recommended daily maximum dose of 80mg was reduced in 2013 by the American Heart Association (ACC/AHA) to 40mg per day [5].

The target location of SV is a reversible inhibition of the HMG-CoA reductase within the cholesterol biosynthesis and mevalonate pathway. The inhibition leads to a reduced intracellular concentration of mevalonate which serves as a regulator of the HMG-CoA reductase. Additionally, the expression of LDL receptors is upregulated [6] Due to the upregulation of the LDL receptor expression the cellular intake of LDL from the circulating blood is increased causing a lowering of the LDL within the circulating blood [6].” (line 25-46)

“Especially the effect of simvastatin upon mineralization was recognized in observational studies which showed beneficial effect of statins on increased bone mineral density (BMD) in subjects affected by osteoporosis [25]. Compared to bisphosphonates statins also influence the mevalonate pathway by inhibiting the HMG-CoA reductase [25]. Beside these observational studies, both in vitro [23,24,26,27,] and in vivo studies [28] proved mineralization effects of SV to bone related cells like osteoblasts.” (line 91-96)

The statins should be introduced better and thorough about theirs effect on tissue mineralization in order to clarify the rationale of the study. Different previous studies described those effect on bone tissue.

The action mechanism and additional information regarding the statins have been added and further information regarding the connection to the odontoblast-like cells have also been provided. As a consequence the reference list has been extended.

“1.1. General function of Statins

Beside others, increased lipid levels in blood could be responsible for first or recurrent coronary events [1]. Since the late 1980s the drugs of the group of statins were administered as a common therapeutic agent to reduce the concentration of low-density-lipoproteins (LDL) in blood [2, 3]. The former used drug to lower the blood concentration of triglycerides was the the so called fibric acid derivatives (fibrates) but the fibrates only show little effect on the circulating LDL blood concentration [4].

The group of statins includes Simvastatin (SV), the whole group of the statins is characterized to act as a 3-hydroxy-3-methylglutaryl coenzyme A (HMG-CoA) reductase inhibitor [3], which lowers the circulating LDL concentration in blood [3]. Out of the group of statins, Simvastatin (SV) is one of the first HMG-CoA reductase inhibitors [3]. During the use over years different types of statins are known and used depending on the diagnosis. Upon often reported rhabdomyolysis a statin related dose-dependent increased risk was supposed. In the following, the formerly recommended daily maximum dose of 80mg was reduced in 2013 by the American Heart Association (ACC/AHA) to 40mg per day [5].

The target location of SV is a reversible inhibition of the HMG-CoA reductase within the cholesterol biosynthesis and mevalonate pathway. The inhibition leads to a reduced intracellular concentration of mevalonate which serves as a regulator of the HMG-CoA reductase. Additionally, the expression of LDL receptors is upregulated [6] Due to the upregulation of the LDL receptor expression the cellular intake of LDL from the circulating blood is increased causing a lowering of the LDL within the circulating blood [6].” (line 25-46)

“Especially the effect of simvastatin upon mineralization was recognized in observational studies which showed beneficial effect of statins on increased bone mineral density (BMD) in subjects affected by osteoporosis [25]. Compared to bisphosphonates statins also influence the mevalonate pathway by inhibiting the HMG-CoA reductase [25]. Beside these observational studies, both in vitro [23,24,26,27,] and in vivo studies [28] proved mineralization effects of SV to bone related cells like osteoblasts.” (line 91-96)

Methods

No details about the age range of the donors has been indicated. Why the teeth were extracted? Were they germs, impacted teeth or erupted ones?

The following explanations were included:

“The study evaluated the effects of SV on 24 primary human odontoblast-like cell cultures originated from different human donors ranging in age from 18 to 35 years. Extraction of the impacted and displaced third molars of all subjects was performed due to lack of space within the dental arch.” (line 302-305)

Results:

Figure 2,3 and 4 presented too low small fonts and should be improved in quality. The bars and line legends should be described in the captions (standard deviation? Minimum/maximum? Mean/Median?)

The figures have been altered. The legends of the bars and lines have been added to the legend of each figure.

“; (the upper and lower whiskers shows the maximum and minimum values, the box shows 50% of the values and the horizontal line within the box shows the median value).” (line 132-133, 142-144, 171-173)

No images of alizarin red colture has been added? Why?

Images of alizarin S red staining were added as a new figure 5 as a representative staining prior to the dilution.

Figure 5. Alizarin S red staining prior to the dilution. From left to right ascending time in days; top row control and below the ascending concentrations as marked on the left side” (line 174-176)

Discussion

Line 99: why the author analyzed osteogenic markers? What is the connection with odontoblasts activity? It should be clarified and discussed.

The connection of ALP conversion as an osteogenic marker is also an indication for odontoblastic activity. Spoto et al. found in 2001 a slight activity of ALP within normal dental pulps with no sign of inflammation. Upon switch to inflammatory stage an increased mineralization activity was evident. Parallel to the increased mineralization the activity of ALP also increased. Thus, a connection seems to be evident.

The additional information has been added to the manuscript within the discussion section.

“Additionally a slight ALP activity was found within inflammatory free healthy dental pulp tissue but the ALP level increase when the pulp status switched to an inflammatory stage,in as far as an increased mineralization with an increased ALP activity were found [44].” (line 231-244)

Discussion:

Line 170: The following sentence is a repetition of introduction section. Conometric retention does not incorporate cement or screw retention. The discussion and conclusion provided a limitate overview due to the lack of studies identified with the electronic databases screening. Probably a deeper manual search could be useful to increase the manuscripts eligible for the review in order to give strength to the discussion.

This comment does not appear to apply to the current manuscript, since no studies on implants have been performed.

Conclusion: The conclusions are generic and should be incentrated to the high survival rate and prosthetic retention capability of the conometric implant systems emerged by the review.

This somment does not appear to apply to the current manuscript, since no studies on implants have been performed.

Round 2

Reviewer 1 Report

.

Reviewer 2 Report

The authors have endeavored to respond to my criticisms, mostly in a satisfactory way.    Regarding point 2 from previous report (MTT test), the author should keep in mind that tetrazolium dyes based assay relies on the cellular metabolic activity of cells. Once cells differentiated they also alter their metabolic function and in turn alter in tetrazolium reduction. Cells with low metabolism reduce very little MTT. I would suggest the authors to consider alternative methods for the future work. 
The author could consider to revise following points in the proofread version:

- Figure 1 - scale bar is still missing. 

- Figure 2,3,4 - text and number on the x- and y-axis can be a bit bigger. It will be easier to read. 

- The author can consider to remove following sentence since no data is provided or discussed in the manuscript text. “Cytotoxic effects were determined with the PierceTM LDH Cytotoxicity Assay (ThermoFisher Scientific) (data not shown)” (line 371)

Reviewer 4 Report

All of the corrections required have been made. The paper is hereby publication ready.